# Effect of the Freeze-Dried Mullein Flower Extract (*Verbascum nigrum* L.) Addition on Oxidative Stability and Antioxidant Activity of Selected Cold-Pressed Oils

**DOI:** 10.3390/foods12122391

**Published:** 2023-06-16

**Authors:** Edyta Symoniuk, Zuzanna Marczak, Rita Brzezińska, Monika Janowicz, Nour Ksibi

**Affiliations:** 1Department of Food Technology and Assessment, Institute of Food Sciences, Warsaw University of Life Sciences, Nowoursynowska St. 159c, 02-787 Warsaw, Poland; 2Department of Chemistry, Institute of Food Sciences, Warsaw University of Life Sciences, Nowoursynowska St. 159c, 02-787 Warsaw, Poland; 3Department of Food Engineering and Process Management, Institute of Food Sciences, Warsaw University of Life Sciences, Nowoursynowska St. 159c, 02-787 Warsaw, Poland; 4Faculty of Sciences of Tunis, University of Tunis El Manar, El Manar I, Tunis 2092, Tunisia; 5Center of Biotechnology of Borj Cedria, Laboratory of Aromatic and Medicinal Plants (LPAM), P.O. Box 901, Hammam-Lif 2050, Tunisia

**Keywords:** antioxidant activity, cold-pressed oils, mullein, oxidative stability, total phenolic content

## Abstract

The aim of the study was to analyze the influence of mullein flower extract addition on the oxidative stability and antioxidant activity of cold-pressed oils with a high content of unsaturated fatty acids. The conducted research has shown that the addition of mullein flower extract increases the oxidative stability of oils, but its addition depends on the type of oil and should be selected experimentally. In rapeseed and linseed oil, the best stability was found for samples with 60 mg of extract/kg of oil, while in chia seed oil and hemp oil, it was found with 20 and 15 mg of extract/kg of oil, respectively. The hemp oil exhibited the highest antioxidant properties, as evidenced by an increase in the induction time at 90 °C from 12.11 h to 14.05 h. Additionally, the extract demonstrated a protective factor of 1.16. Oils (rapeseed, chia seed, linseed, and hempseed) without and with the addition of mullein extract (2–200 mg of extract/kg of oil) were analyzed for oxidative stability, phenolic compounds content, and antioxidant activity using DPPH^•^ and ABTS^•+^ radicals. After the addition of the extract, the oils had from 363.25 to 401.24 mg GAE/100 g for rapeseed oil and chia seed oil, respectively. The antioxidant activity of the oils after the addition of the extract ranged from 102.8 to 221.7 and from 324.9 to 888.8 µM Trolox/kg for the DPPH and ABTS methods, respectively. The kinetics parameters were calculated based on the oils’ oxidative stability results. The extract increased the activation energy (Ea) and decreased the constant oxidation rate (*k*).

## 1. Introduction

Cold-pressed oils have gained popularity among consumers due to their higher nutritional value compared to commonly used refined oils. However, oils high in polyunsaturated fatty acids are susceptible to low oxidative stability and short shelf life [1]. The deterioration of food, including lipid changes, is caused by chemical and enzymatic reactions, which can lead to undesirable sensory properties such as unpleasant taste and smell, as well as loss of nutritional value [2]. The rate of oxidation is influenced by various factors, including fatty acid composition, presence of antioxidants and pro-oxidizing compounds, and external factors such as storage time, temperature, radiation, oxygen exposure, and light [3]. While the chemical composition of oils can be regulated to some extent, external factors can be controlled at every stage of production, from extraction to storage. Therefore, it is crucial to implement various processes and barriers to protect the oil from external influences and prevent rapid oxidation [4].

Oxidation can lead to the formation of harmful products that may affect the safety and health of consumers. As a result, food technologists commonly use both natural and synthetic antioxidants to extend the shelf life of food products [5]. Recent studies have shown that natural antioxidants can be obtained using environmentally friendly and unconventional extraction methods and that edible oils enriched with natural antioxidants can exhibit comparable or higher antioxidant activity and thermal stability compared to synthetic antioxidants. However, it is important to determine the optimal concentration and combination of antioxidants and explore the potential synergistic effects of natural antioxidants [6]. Natural antioxidants can be derived from various plant parts, including flowers. Mullein flower (*Verbascum* L.) extract could be an example of such an additive [7].

Mullein is a medicinal plant that has been used for centuries in traditional medicine and is widely cultivated for its numerous health benefits [8]. The flowers of mullein contain flavonoids, iridoids and their glycosides and triterpene saponins, phytosterols, triterpenes, polysaccharides, phenolic acids, and their glycosides, as well as essential oils, which contribute to its valuable composition and various therapeutic effects such as antiviral, analgesic, anti-spasmodic, expectorant, hepatoprotective, and cardiovascular strengthening properties [9]. Furthermore, mullein flowers have been traditionally used for lung disorders, infections, hemorrhages, diuretic effects, and cough syrups and as relaxants for lesions, skin complications, swelling, and inflammatory disorders of the respiratory tract, among others [10,11]. Saponins, which are secondary metabolites present in the form of triterpenoids, are known to exhibit surface-active properties and have potential applications as natural emulsifiers, foaming agents, stabilizers, and drug delivery agents in the food and pharmaceutical industries [12]. In fact, saponins have been studied for their efficient additive effects in the formation and stabilization of oil-in-water and nano-emulsions [13]. Furthermore, studies have demonstrated the beneficial effects of saponins on health, including antifungal and bactericidal properties. Verbascoside, a phenylethanoid glycoside, has been identified as a major contributor to the antioxidant capacity of *V. nigrum* extract. Flavonoids have been attributed to the diuretic effects of *V. nigrum* extract [8,14].

Despite the numerous health-promoting and medicinal properties of mullein, there is limited research on the use of mullein extracts as natural antioxidants, but the results of various studies show that mullein flowers have antioxidant potential. The antioxidant activity of mullein extracts is primarily related to the presence of various flavonoid compounds such as chlorogenic, vanilic, caffeica, cinnamic acid, verbascoside, and harpagoside [8,11]. Therefore, this study aims to investigate the effect of adding freeze-dried mullein flower extracts on the antioxidant properties and oxidative stability of cold-pressed oils, particularly those with a high content of polyunsaturated fatty acids that are susceptible to oxidation, while also leveraging the medicinal properties of mullein.

## 2. Materials and Methods

### 2.1. Preparation of Research Material

The research materials used in this study comprised four cold-pressed oils: rapeseed oil (RO), hempseed oil (HO), linseed oil (LO), and chia seed oil (CHO). These oils were obtained from the manufacturers immediately after pressing, with the oil temperature at the press outlet maintained at around 38 ± 2 °C. Throughout the research period, the oils were stored in their original dark glass bottles under refrigeration conditions. Additionally, freeze-dried mullein (*Verbascum nigrum* L.) flowers were used as an additive (Figure 1). The flowers were harvested from a meadow near the Kozienicka Forest (51°49′ N, 21°51′ E), Mazovian Voivodeship, Poland, in September 2021. The flowers were harvested during the beginning of the flowering period, when they were yellow, around noon when the flowers were fully open. The harvested material was frozen in a ProfiMaster PMU0452 shock freezer (GmbH, Germany) at −40 °C for at least 5 h and then freeze-dried in an Alpha 1–4 LSC plus device (Christ, Germany) for 48 h at a shelf temperature of 30 °C and pressure of 63 Pa. The test samples were prepared by mixing each of the oils with different concentrations of alcoholic extract, in the form of syrup according to Máriássyová [15] (Figure 1) (2, 5, 10, 15, 20, 30, 40, 60, 100, and 200 mg of extract per 1 kg of oil), as well as by using fresh oil without the addition of extract. For each oil, a sample was selected for which the protection factor was calculated based on oxidative stability. All analyses were performed before the end of the data validity, within two weeks. During this time, the oils were stored under refrigerator conditions at 4 °C, and the extracts were stored in the freezer at −18 °C.

### 2.2. Reagents

All reagents and solvents used for the analysis and sample preparation for gas chromatography (GC) were of high-performance liquid chromatography/gas chromatography (HPLC/GC) purity and were obtained from POCH S.A. (Gliwice, Poland). The Food Industry 37 Component Fatty Acid Methyl Esters (FAME) mix standard was supplied by Restek (Bellefonte, PA, USA). Distilled water with a conductivity of 0.05 µS was obtained using an HLP Smart 2000 apparatus (Hydrolab, Straszyn, Poland). Other reagents and standards were purchased from Merck (Darmstadt, Germany).

### 2.3. Oils’ Quality Assessment

Fatty values, including acid value (AV), peroxide value (PV), and p-anisidine value (*p*-AnV), were analyzed to assess the initial quality of the oils and their quality after the addition of mullein extracts. The methods used for analysis were AOCS Cd 3d-63 [16] for AV, AOCS Cd 8–53 [17] for PV, and AOCS Cd 18–90 [18] for *p*-AnV. The total oxidation value, also known as the TOTOX indicator, was calculated using the formula: TOTOX indicator = (2 × PV) + *p*-AnV.

### 2.4. Fatty Acids Composition Determination

The fatty acid composition of the cold-pressed oils was determined using gas chromatography, following the AOAC method 996.06 [19]. Methyl esters of the oil samples were prepared by dissolving 0.1 g of the oil in 2 mL of hexane and 0.5 mL of 2 M methanolic KOH. After mixing and phase clarification, 1 mL of the hexane phase was collected in a vial. The fatty acid profile was determined using a Thermo Scientific Trace 1300 gas chromatograph equipped with an FID (flame ionization detector), using an SGE BPX70 high-polar capillary column for separation. The initial temperature during analysis was set at 100 °C for the first four minutes and then raised to 240 °C at a rate of 3 °C/min. Helium was used as the carrier gas with a flow rate of 40 mL/min. A standard mixture of 37 FAME methyl esters from Restek was used for identifying the fatty acids, and the results were reported as percentage content.

### 2.5. Preparation of Ethanolic Mullein Flowers Extracts

The ethanolic extract of mullein flowers was obtained through three steps of alcoholic extraction. Initially, freeze-dried mullein flowers were ground using a mortar. The first step involved extracting 2 g of flowers with 40 mL of 60% ethanol at 60 °C for 1 h in a water bath with shaking (Vibra 6, AJL, Cracow, Poland). Following the extraction, the sample was centrifuged for 5 min at 5000 rpm using a centrifuge (MPW-352, MPW, Poznań, Poland). The resulting extract was transferred to another vessel, and the residue underwent a second extraction step using 30 mL of 60% ethanol, following the same parameters and post-treatment as the first stage. The third extraction step utilized 30 mL of solvent, with the procedure remaining consistent with the first and second extraction steps. Once the extractions were complete, the extracts were collected, filtered, and evaporated in a vacuum evaporator (Rotavapor R-210, Buchi, Flawil, Sankt Gallen, Switzerland) at 60 °C, resulting in a final volume of 30 mL for the ethanolic extract.

### 2.6. Methanolic Extract Preparation for Bioactive Compounds Determination

Prior to the determination of bioactive compounds, methanolic extracts of oils were prepared. A sample of the oil or oil with the flower extract (3 g) was dissolved in 15 mL of hexane and subjected to three consecutive extractions with methanol (3 × 5 mL) using vortexing. After stratification, the methanol fraction was washed with hexane to remove residual oil.

### 2.7. Total Phenolics Content Determination

The total content of phenols in the cold-pressed oils was determined using the Folin–Ciocalteu (F–C) reagent, according to the method described by Dewanto et al. [20]. A total of 0.125 mL of the extract was mixed with 0.5 mL of distilled water and 0.125 mL of the Folin–Ciocalteu reagent. After 3 min, 1.25 mL of 7% (*m*/*v*) sodium carbonate solution was added and made up to 3 mL with distilled water. After 1.5 h, the absorbance of the samples was measured at a wavelength of λ = 760 nm. The quantitative determinations’ results (total phenolic content—TPC) were presented as gallic acid equivalent (mg GAE/100 g of oil). Absorbance was measured using the Spectrophotometer UV-Vis GENESYSTM 180 Thermo Scientific, Waltham, MA, USA.

### 2.8. Determination of the Total Antioxidant Capacity by the Reduction in DPPH Free Radical

The total antioxidant capacity of the oil extracts was determined using the DPPH free radical assay, following the methodology described by Pająk et al. [21]. In this method, 0.1 mL of the oil extract in methanol was added to 3.9 mL of DPPH solution. After one hour, the absorbance of the solution was measured at a wavelength of 515 nm. The concentration of DPPH in the reaction medium was determined using a calibration curve prepared with standard Trolox solutions (TEAC). Absorbance was measured using the Spectrophotometer UV-Vis GENESYSTM 180 Thermo Scientific.

### 2.9. Determination of Total Antioxidant Capacity by ABTS Radical Cation Reduction Method

The total antioxidant capacity of the oil extracts was determined using ABTS cation radicals (3-ethylbenzothiazoline 6-sulfonic acid) following the method of Czeniakowska-Szydło and Łaszewska [22]. For the antioxidant capacity determination, 0.2 mL of the oil extract was added to 3.5 mL of the ABTS solution, and the mixture was shaken and incubated for 6 min. The radical cation was prepared by dissolving ABTS and potassium persulfate in methanol, and the initial absorbance was set to 700. The solution was then stored in the dark for 24 h. The absorbance of the reaction mixture and ABTS radicals was measured at λ = 734 nm. The exact concentration of ABTS in the reaction medium was determined from a calibration curve using standard Trolox solutions (TEAC). The absorbance value of the reaction mixture before adding the oil extract was recorded and used to calculate the antioxidant capacity. Absorbance measurements were performed using the Spectrophotometer UV-Vis GENESYSTM 180 Thermo Scientific.

### 2.10. Determination of Oxidative Stability Using the Rancimat

The oxidation stability of the oil samples was measured using a Rancimat type 892 apparatus by Metrohm (Herisau, Switzerland). Samples of fresh oil weighing 3 g, with and without the addition of extract, were placed in a heating block, and a stream of air at a rate of 20 L/h was passed through the sample. The volatile oxidation products formed during the measurement were carried along with the air stream to the measuring vessel, which contained 60 mL of deionized water. Measurements were carried out at four temperatures: 90, 100, 110, and 120 °C for linseed, chia seed, and hempseed oil, and at 110, 120, 130, and 140 °C for rapeseed oil. The determination of oxidative stability for each tested oil was performed in two parallel repetitions, and the arithmetic mean of the two results was reported as the final value.

### 2.11. Determination of the Oxidation Kinetics Parameters

The oxidation induction times determined with the Rancimat apparatus were used for the calculations. Calculations were performed according to Farhoosh et al. [23]. Based on the obtained results, graphs were prepared on a semi-logarithmic scale, with the dependence of the decimal logarithm of the induction time on temperature and the reciprocal temperature.

The regression lines were determined according to the following equation:lnτRancimat = 1/T,(1)
where lnτRancimat—natural logarithmic induction time determined in the Rancimat and T—oxidation temperature [K].

Using the obtained results and the Arrhenius formula:k = Zexp (−Ea/RT),(2)

The basic parameters of the oil oxidation kinetics were calculated: activation energy (Ea), pre-exponential factor (Z), and reaction rate coefficient (k) at measurement temperatures.

The enthalpy (ΔH) and entropy (ΔS) were calculated according to Cherif et al. [24] based on the equation derived from the activated complex theory:log(k/T) = log(k_B_/h) + (ΔS/R) − (ΔH/RT),(3)
where k_B_—Boltzmann constant (1.380649 × 10^−23^ J/K), h—Planck’s constant (6.6260755 × 10^−34^ Js), R—gas constant (8.314 J/mol K),
and regression line: logk/T = 1/T,(4)
where logk—decimal logarithm of the reaction rate coefficient (k) and T—oxidation temperature [K].

The Gibbs free energy (ΔG) takes into account both the enthalpy (heat energy) and entropy (disorder or randomness) of a system. The Gibbs free energy was calculated according to the formula:ΔG = ΔH − T × ΔS(5)

The Q10 parameter was also calculated according to Equation (6), and based on this parameter, the induction times of oils at temperatures of 25 and 4 °C were calculated.
Q10 = (R_2_/R_1_)^(10/(T^_2_^−T^_1_^)^(6)

### 2.12. Statistical Analysis

All experiments were carried out in triplicate for each oil. Statistica version 13.3 (StatSoft, Inc., Tulsa, OK, USA) was used to analyze the obtained experimental results. One-way analysis of variance (ANOVA) and Tukey’s test were performed with a *p*-value ≤ 0.05 for all chemical analyses, and for kinetic parameters, the *p*-value was equal to 0.001. In addition, principal component analysis (PCA) was conducted to explore the relationship between induction time and chemical composition, as well as induction time and kinetics parameters. The cluster analyses were used to group the oils according to the data taken into account for PCA analyses.

## 3. Results and Discussion

### 3.1. Effect of the Addition of Mullein Flower Extract on Oxidative Stability of the Tested Oils

Oxidative stability is a critical indicator of the quality and safety of cold-pressed oils, as it reflects their susceptibility to oxidation and predicts their shelf life. However, the oxidation process can lead to the formation of toxic compounds that degrade the sensory characteristics and nutritional value of the oil.

To assess the effect of an extract on the stability of oils, the study measured the oxidative stability of oils both with and without the addition of the extract. The coefficient of variation was also determined, and the protective factor of the extract was calculated (Table 1).

The results obtained from the oxidative stability analysis of rapeseed oil at 110 °C (Table 1) suggest that the addition of mullein extract increased its oxidative stability. The rapeseed oil with an addition of 60 mg of mullein extract per kilogram of oil exhibited the longest induction time, indicating the best oxidation stability of 10.79 h, with a protection factor of 1.10. This result was almost an hour longer than the rapeseed oil without mullein addition, and the induction time was extended by 10%. On the other hand, the shortest induction time, i.e., the worst oxidative stability, was observed in the oil without mullein flower extract, which was 9.81 h. In contrast, the longest induction time was observed in hemp oil with a concentration of 15 mg of extract per kilogram of oil, which was 14.05 h. Therefore, it can be concluded that a lower concentration of mullein extract is required to increase the stability of hemp oil compared to the other tested oils. Furthermore, the increase in the concentration of the extract in all oils initially resulted in an increase in the induction time up to a specific concentration, after which it decreased. This may be attributed to the type, isomer, and concentration of antioxidant compounds, as well as the possible synergistic action of bioactive compounds in the oil and extract. It is known that edible oils often contain multiple-component antioxidants that can interact with each other. Consequently, excessively high concentrations of antioxidants may have an adverse effect on the oxidative stability of oils [3].

The hemp oil without the addition of mullein extract exhibited the shortest induction time of 12.11 h. Statistical analysis revealed two homogeneous groups, with most of the ten samples showing significant differences from the control (hemp oil without mullein flowers) with a *p*-value ≤ 0.05. The protective coefficient (PF) of hemp oil showed the least variability among the concentrations, ranging from 1.12 to 1.16. Mullein extract had the most pronounced effect on the oxidative stability of hemp oil, with an increase in induction time by 16% compared to other oils tested. Similarly, in the case of rapeseed oil, the longest induction time was observed in linseed oil with the addition of 60 mg of extract per kilogram of oil, which was 7.25 h. However, the protection factor was lower at 1.08. The difference between the sample with the longest induction time and the control sample without mullein flower extract was approximately half an hour, similar to rapeseed oil. The shortest induction time in linseed oil was obtained with a sample containing 2 mg of extract per kilogram of oil, which was 6.52 h. The results for linseed oil were not statistically differentiated, with only two homogeneous groups identified at a significance level of *p*-value ≤ 0.05, and no sample differing significantly from the sample without mullein flower extract. The protective factor (PF) of linseed oil ranged from 1.00 to 1.08, indicating a moderate level of protection against oxidation, with the highest PF values observed in samples containing 40 and 60 mg of extract per kilogram of oil. On the other hand, chia seed oil showed the lowest oxidative stability among all the oils tested, with an induction time not exceeding 7 h. Among the chia seed oil samples, the longest induction time of 6.96 h was observed in a sample with 20 mg of extract per kilogram of oil, which was statistically different from the control sample without extract. The shortest induction time was 6.44 h. Similar to other oils, chia seed oil exhibited limited variability in PF, ranging from 1.02 to 1.08.

The obtained protective factor (PF) results for the analyzed oils were higher than those reported by Turan [25] for rapeseed oil with the addition of bay, sage, and thyme extracts, which were from 1.09 to 1.10, from 1.04 to 1.09, and 1.09 (at 200 mg/kg), respectively. Moreover, the PF values for the addition of BHA (0.99) and BHT (1.04) at a concentration of 200 mg/kg of oil were lower than those obtained in our study. Additionally, the addition of mullein extract showed a superior effect on the oxidative stability of hemp and linseed oil compared to lemon balm or oregano extract, as reported by Nikolov et al. [26]. However, rosemary and lavender extracts showed better results in terms of PF according to other plant sources, indicating a relatively high antioxidant potential of mullein extract.

### 3.2. Impact of the Extract Addition on the Basic Distinguishing Features of Oil Quality

Based on the oils’ oxidative stability and protection factor results, samples were selected for the extract’s impact on the primary highlight of oil quality. The impact of mullein flower extracts on oils’ basic quality features was presented in Table 2.

According to Codex Alimentarius [27], the acid value for cold-pressed oils should not exceed 4 mg KOH/g of the sample, and all the analyzed oils in our study met this requirement. The lowest acid value was observed in hemp oil (HO) at 1.01 mg KOH/g, while the highest acid value was found in linseed oil with the addition of mullein (LOM) at 2.23 mg KOH/g. Only in the case of chia seed oil, the addition of mullein extract did not have a statistically significant effect on the acid value. However, in all other cases, the samples with the addition of mullein extract showed statistically significant differences compared to the samples without the extract. Ratusz and Wirkowska [28], in their study on the effect of adding antioxidants to rapeseed, soybean, and sunflower oil, noticed a slight increase in the acid value in two out of three tested oils. Conversely, in the studies by Krajewska and Kachel [29], the addition of herbs reduced the acid value of black cumin, hemp, and linseed oil. The slight increase in the acid value observed in our study may be attributed to the addition of an extract containing water, which could have caused partial hydrolysis of the oils.

The results of peroxide value determination revealed more significant discrepancies compared to the acid value. Hemp oil exhibited much higher values compared to other oils. Despite this, all oils were of good quality, with values below the limit of 15 mEq O_2_/kg set by Codex Alimentarius [27] for cold-pressed oils. The lowest peroxide value was observed in chia seed oil, both with and without the extract, at 0.77 mEq O_2_/kg, while the highest peroxide value was found in hemp oil with the addition of mullein flower extract at 10.50 mEq O_2_/kg. Rapeseed, linseed, and chia seed oils showed similar peroxide values with and without the extract, and there was no statistically significant difference between them at a significance level of *p*-value ≤ 0.05. However, in the case of hemp oil, the obtained results differed significantly, which could be attributed to the high content of chlorophylls in hemp oil, which can act as prooxidants upon exposure to light. Phenolic compounds present in the extract may have reacted with radicals and formed peroxide structures during analysis, leading to higher titration results [3]. Similar tendencies were observed by Ratusz and Wirkowska [28] in their research. They found that the addition of antioxidants caused an increase in the peroxide value, with the most significant increase observed in rapeseed oil, where the peroxide value increased by 65% compared to the sample without the antioxidant. Spano et al. [30] also reported similar results for hemp oil from a local producer, with peroxide values exceeding 20 mEq O_2_/kg. On the other hand, Aladić et al. [31] obtained a much lower content of primary oxidation products compared to the tested hemp oil, with a peroxide value of 3.9 mEq O_2_/kg. Furthermore, despite an increase in peroxide value following the addition of the antioxidant, the overall stability of the oils improved. This can be attributed to the fact that at higher temperatures, such as 80 °C, the formation of hydroperoxides occurs alongside their decomposition at a notable rate. These hydroperoxides break down into both volatile and non-volatile compounds, with the Rancimat apparatus focusing on volatile oxidation products to determine the outcome. Additionally, the extract contains chain-breaking antioxidants that function by capturing peroxyl radicals through oxidation reactions. Although the peroxide values may be higher, the presence of antioxidants can extend the oxidation time [32].

The results of the determined p-anisidine values showed significant diversification, indicating varying levels of secondary oxidation products in the tested oils. The obtained p-anisidine values for oils without extract were low, which is typical for cold-pressed oils, as high temperatures are not used during their production. The lowest value was observed in chia seed oil with mullein extract at 0.19, while the highest value was found in linseed oil with extract at 1.71. The most significant difference in the anisidine value was observed between oil without extract and oil with its addition in linseed oil. However, higher content of secondary oxidation was detected for all analyzed oil. The increase in the content of secondary oxidation products in oils may be related to the determination method, which involves the detection of compounds with a carboxyl group. One such compound formed as a result of oxidation reactions in oils is mainly malonic acid. Phenolic compounds present in the extract may contain carboxyl groups that react with p-anisidine, leading to higher test results. Additionally, these differences may be due to the fact that the method is based on absorbance determination, which can pose challenges in the analysis of cold-pressed oils with varying color intensities. Only linseed oil showed statistically significant differences at a significance level of *p*-value ≤ 0.05, while the remaining oils belonged to the same homogeneous group for oils without mullein extract and with its addition.

The Totox indicator is a parameter used to assess the overall extent of fat oxidation. Hemp oil showed significant differences compared to other oils, while the remaining oils were below the maximum limit specified in the standards. Hemp oil without extract had a Totox indicator value of 13.34, and with the addition of extract, it increased to 22.20, exceeding the permissible value. Among the other oils, the lowest Totox indicator value was observed in chia seed oil with mullein flower extract. Rapeseed and chia seed oils did not show statistically significant differences compared to their counterparts with mullein extract. However, there were variations in the Totox indicator values for linseed oil and hemp oil.

### 3.3. Impact of the Mullein Addition on the Composition of Fatty Acids Tested Oils

The fatty acid composition of oil plays a crucial role in determining its oxidative stability. Unsaturated fatty acids are more susceptible to oxidation compared to saturated fatty acids, as they contain double bonds that can react with oxygen in the air. Among unsaturated fatty acids, polyunsaturated fatty acids (PUFAs) are particularly prone to oxidation due to their multiple double bonds, which make them highly reactive. Monounsaturated fatty acids (MUFAs) are more stable than PUFAs but less stable than saturated fatty acids. The fatty acid composition of the analyzed oils, both with and without the extract, with the highest PF, was shown in Table 3.

The fatty acid composition of the analyzed oils was found to be typical for the raw material used, and the addition of freeze-dried mullein extract did not alter the fatty acid composition [33,34,35,36]. Rapeseed oil is generally considered to be a stable oil, particularly when compared to linseed oil, due to its high content of monounsaturated fatty acids and low content of saturated fatty acids, which contribute to its stability. Chia and linseed oil had the highest polyunsaturated fatty acid (PUFA) content among the analyzed oils, with chia containing 87% and linseed containing 76–77% of PUFAs. Both oils were also rich in α-linolenic acid (α-C18:3), with chia containing 69% and linseed containing 65%. Chia oil had the lowest levels of saturated fatty acids (6%) and monounsaturated fatty acids (7.5%). Literature data suggest that linseed oil was of a high linolenic variety, as it had a very high content of linolenic acid. Hempseed oil was the only oil in which Ɣ-linolenic acid (Ɣ-C18:3) was detected, at a level of 3.5%. Furthermore, hempseed oil had the highest linoleic fatty acid (C18:2) content at 56–58% and a monounsaturated fatty acid content similar to linseed oil.

The fatty acid composition of an oil plays a significant role in its oxidative stability. Oils that have a higher proportion of polyunsaturated fatty acids (PUFAs) are more prone to oxidation compared to oils that primarily contain monounsaturated or saturated fatty acids. Among the analyzed oils, chia seed oil had the lowest oxidative stability and the highest content of polyunsaturated fatty acids (~87%). Linseed oil was slightly more stable, but the difference in the content of polyunsaturated fatty acids between linseed oil and chia oil was not statistically significant. The slightly longer induction time observed in linseed oil may be due to its higher content of C18:1 acid and lower content of C18:2. It is worth noting that rapeseed oil was analyzed at a different temperature (110 °C) than the other oils, and the obtained result was approximately 11 h. At a lower temperature (90 °C), this time would be four times longer. As a result, rapeseed oil exhibited the highest stability among the tested oils, which can be attributed to its fatty acid composition.

### 3.4. Impact of the Extract on the Total Phenol Content and Antioxidant Activity

The total phenolics content is closely related to the antioxidant activity in the tested oils. Among all oils without the addition of mullein, hempseed oil was characterized by the highest content of phenols—394.64 mg GAE/100 g oil—and the lowest—rapeseed oil with a content of 235.42 mg GAE/100 g of oil (Table 4).

The addition of mullein flower extract significantly influenced the content of phenolics compounds. The lowest content of phenols was detected for rapeseed oil and chia seed oil without the addition of mullein extract (235.42 and 313.70 mg GAE/100 g oil, respectively). The content of phenols increased with increasing the addition of the extract. Oils with the highest amount of extract addition—200 mg/kg of oil—were characterized by the phenol content from 372.18 to 476.96 mg GAE/100 g. The situation is similar in the case of linseed and hemp oil. Only samples of linseed oil with 100 and 200 mg of extract/kg significantly increased phenolics content. As a result, when comparing the total phenolics of the four oils with the maximum amount of extract added, linseed oil still has the highest value, and rapeseed oil has the lowest, amounting to 476.96 and 372.18 mg GAE/100 g of oil, respectively. Statistical analysis showed 11 homogeneous groups at the significance level of *p*-value ≤ 0.05, which indicates the high variability of the results obtained and the significant influence of the additive mullein. Kozłowska and Ścibisz [37] studied the content of phenolic compounds in spices extracts and showed that plants such as thyme, oregano, mint, and sage have a high content of phenolic compounds. Spiridon et al. [38] studied extracts from 10 different herbs, including Verbascum, and they obtained a result of 45 mg GAE on a gram of extract. Alizadeh [39] determined the total content of phenols in a rare species of the Satureja rechingeri plant, ranging from 355 to 375 mg GAE/100 g. Parry et al. [40] examined the total content of phenolic compounds in mullein seed oils and obtained a 180 mg GAE/100 g oil. This result is significantly lower than that obtained, probably because the tested oil was oil from mullein, not oil with the addition of an extract, as in this case.

The present study evaluated the four cold-pressed seed oils for their scavenging activity against the ABTS^•+^ radicals generated by chemical oxidation reactions. All oil extracts directly reacted with and quenched ABTS^•+^, with an AA range of 171.1–804.2 µmol TE/g of oil and of 324.9–888.8 µmol TE/g of oil without and with mullein extracts, whereas the greatest ABTS^•+^ scavenging capacity was observed in the cold-pressed hempseed oil (Table 5). Oils were also analyzed for their antioxidant activity using the DPPH^•^ radical method. According to the obtained results, mullein extract affects oils’ antioxidant activity. The best antioxidant properties determined using the DPPH^•^ radical showed hemp oil (147.8 µmol TE/kg), while the worst was chia seed oil (102.8 µmol TE/kg). After adding the extract for the samples with the best protective factor from the analysis of oxidative stability, it was found that the best antioxidant properties had cold-pressed rapeseed oil with the addition of 60 mg/kg mullein extract (221.7 µmol TE/kg). Results obtained for oils without the extract addition were similar to those presented by Prescha et al. [41].

### 3.5. Parameters of Oxidation Kinetics of Analyzed Oils

The stage preceding the calculation of the oxidation kinetics parameters was the analysis of the stability of oils without addition and with the addition of mullein flower extract at various oxidation temperatures. The obtained results of the analysis of the oxidative stability of oils are presented in Table 6. Analysis of the oxidative stability of oils with and without the extract addition showed that the induction times of the oils were statistically significantly different. At each of the analyzed temperatures, the induction time of oils with the addition of the extract was longer than those without additives. As the temperature increased, the induction time was shorter in all cases. The most significant differences in oil induction times were observed at the lowest temperature of −80 °C.

Based on the determined induction times using the Rancimat apparatus, plots were made on a semi-logarithmic scale of the logarithm of induction time versus temperature and the logarithm of induction time versus the reciprocal of temperature. Obtained equations were used to calculate the parameters of the oxidation kinetics (Table 7).

The activation energy (Ea) is one of the basic parameters of oxidation kinetics, defined as the smallest amount of energy needed to initiate a reaction. According to Ratusz et al. [42], activation energy depends mainly on the fatty acid composition of vegetable oils. A high content of SFA and MUFA is responsible for the high activation energy, and a high content of PUFA is responsible for the low activation energy. The analysis of the fatty acid composition has shown that rapeseed oil has the highest content of MUFA and SFA. In contrast, chia seed oil has the lowest. These results confirmed the thesis of Ratusz et al. [42] because rapeseed oil has the highest activation energy—82.48 and 84.11 kJ/mol. The lowest activation energy was calculated for hemp oil without addition and with mullein extract—76.06 and 77.34 kJ/mol, respectively. The lowest values of the activation energy of the reaction could be associated with the highest values of the discriminants determining the degree of oil oxidation, i.e., the values of the PV and the Totox indicator. All tested oils with the addition of mullein extract were characterized by higher activation energy, indicating harder oxidation reaction initiation. Kozłowska and Żontała [43] investigated the stability of cold-pressed sunflower oil after adding water-ethanol extracts of thyme, marjoram, oregano, basil, savory, peppermint, and sage in the amount of 0.01 and 0.04%. Nearly all of their samples showed a higher activation energy than the control sample without additives, the Ea of which was 85.39 kJ/mol. The remaining samples, with the addition of extracts, ranged from 85.21 (basil 0.01%) to 102.33 kJ/mol (oregano 0.04%).

The constant reaction rate (k) is an important parameter in chemical reactions as it is directly proportional to the speed of the reaction. The study’s results suggest that the rate of oxidation decreases as the temperature decreases. Rapeseed oil, with and without the addition of mullein flowers, had the lowest constant rate of oxidation at 100 °C, while hempseed oil had the highest value for this parameter. The addition of mullein ethanolic extract lowered the value of k for each oil, resulting in a slower oxidation reaction. In a similar study, Symoniuk et al. [44] found that the constant rate of oxidation for five linseed oils was similar in a temperature range of 70–105 °C. Likewise, Romagnoli et al. [45] observed a lower reaction rate for biodiesel (a mixture of oils and animal fat) after the addition of natural antioxidants such as senna leaves, blackberry fruit, and hibiscus flower extracts. The addition of one extract resulted in a slightly slower oxidation rate. Additionally, the obtained results showed that mullein flower extracts had a better rate-slowing effect on the oxidation reaction than thyme, marjoram, or BHA [46].

The Q10 parameter is used to evaluate the effect of temperature on the rate of oxidation reactions, such as lipid oxidation or autoxidation of food compounds. It represents the change in reaction rate or kinetics of an oxidative process with a 10-degree Celsius increase in temperature. The Q10 values of the analyzed oils ranged from 1.81 for rapeseed oil to 1.86 for linseed oil with the addition of mullein flower extract. After adding the extract, the Q10 values slightly increased for the oils with the extract addition. In contrast, Hashemi et al. [47] reported a significant reduction in the Q10 value for oil blends with both synthetic antioxidants and extracts. A higher Q10 value indicates that the oxidation process is more sensitive to temperature changes, while a lower value indicates that the process is less sensitive to temperature changes. The Q10 parameter was used to estimate the shelf life of oils at 25 and 4 °C. The results showed that rapeseed oil had the most favorable shelf life, with a 20% longer storage time than oil without the extract. In contrast, linseed oil and hemp oil had the shortest storage times.

Two basic parameters calculated for oxidation kinetics are entropy (ΔS) and enthalpy (ΔH). ΔS^++^ determines the degree of energy dissipation and disorder of a system, while ΔH^++^ determines the probability of spontaneous reaction occurrence [48]. All tested oils had negative ΔS values, ranging from −123.5 to −134.2 J/(mol K), indicating a greater ordering of active complexes compared to the reactant molecules. Rapeseed oil with mullein had the lowest entropy, and chia seed oil had the highest. However, in this study, the oils with the lowest entropy values, rapeseed oil with and without mullein, exhibited the best oxidation stability in the Rancimat apparatus. A lower entropy value indicates a higher chance of the formation of active complexes, which slows down the reaction [48,49]. Hempseed oil had the lowest enthalpy, indicating a higher probability of spontaneous oxidation initiation, while the highest enthalpy was observed in rapeseed oil. The differences in enthalpy values between oils with and without mullein were minor. The Gibbs free energy (∆G) constant, calculated using these two parameters, was found to be from 121.11 to 126.97 kJ/mol at 100 °C for the tested oils, with oils with added extract characterized by higher values. Higher ∆G^++^ values correspond to lower rates of lipid oxidation reactions and, thus, higher oxidative stability of the lipid system. The increase in the oxidation temperature for each oil slightly amplified this effect. This could be attributed to the endothermic nature of the activated complexes formed during the thermal oxidation process of the oils, as well as the lower disorder rate of the reactants in the activated complexes. In conclusion, positive ∆G values indicate that the lipid oxidation reaction is non-spontaneous.

Principal component analysis (PCA) is a statistical technique used for reducing the dimensionality of a data set while retaining as much of the variability in the data as possible. It works by transforming the data into a new set of variables called principal components, which are linear combinations of the original variables. The data set used in this PCA analysis consists of oils’ chemical composition for a set of induction period, peroxide value, acid value, *p*-anisidine value, Totox indicator, fatty acids composition, antioxidant activity, and total phenolic compounds. The results of the PCA analysis showed that the first principal component accounted for 44.43%, and the second principal component accounted for 27.61% of the variability. The presented results of the PCA analysis illustrate correlations between variables. As shown in the presented biplot (Figure 2), the induction time of oils depended mainly on their fatty acid composition, PUFA and MUFA content, and, to a lesser extent, on the peroxide value and total content of phenolic compounds. Oils were grouped into four groups based on their similarity, although oils with and without extract were found in the same groups, indicating that oils did not differ significantly in terms of their composition and oxidative stability.

As per PCA analysis for induction time and kinetics parameters, it could be concluded that induction time was highly correlated with Ea, ΔH, ΔS, *k*, and Q10 (Figure 3). The data set used in this PCA analysis consists of oils’ kinetic parameters and induction period. The results of the PCA analysis showed that the first principal component accounted for 67.91%. The second principal component accounted for 24.9% of the variability. Oils’ induction time was highly correlated with the constant reaction rate, activation energy, and Gibbs free energy constant. Based on the kinetics of oxidation parameters, the oils were divided into three groups according to their similarity. Rapeseed and hemp oils formed two distinct groups, while chia seed and linseed oils did not differ from each other.

## 4. Conclusions

The study revealed that the addition of mullein flower extract could enhance the oxidative stability and antioxidant activity of oils, with the amount of extract varying depending on the type of oil. For rapeseed and linseed oils, the highest stability was observed with the addition of 60 mg of extract/kg of oil, whereas for chia seed and hemp oils, it was 20 mg/kg and 15 mg/kg, respectively. The antioxidant activity of the oils was found to be directly related to the concentration of the extract, with greater inhibition and activity observed at higher concentrations. Moreover, the addition of mullein extract increased the total phenolic compound content in all oils, with linseed oil having the highest and rapeseed oil having the lowest. Finally, the study demonstrated that adding mullein extract to oils could inhibit their oxidation reaction by increasing their activation energy and reducing the constant rate of oxidation.

## Figures and Tables

**Figure 1 foods-12-02391-f001:**
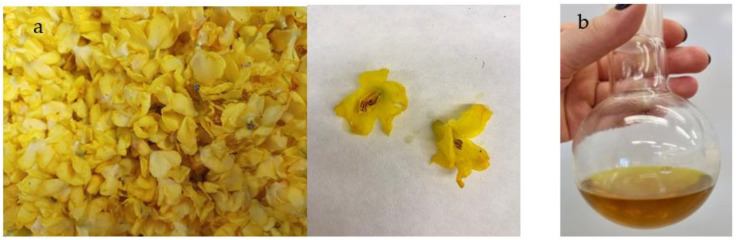
Mullein flowers (*Verbascum nigrum* L.) (**a**) and flower extract (**b**).

**Figure 2 foods-12-02391-f002:**
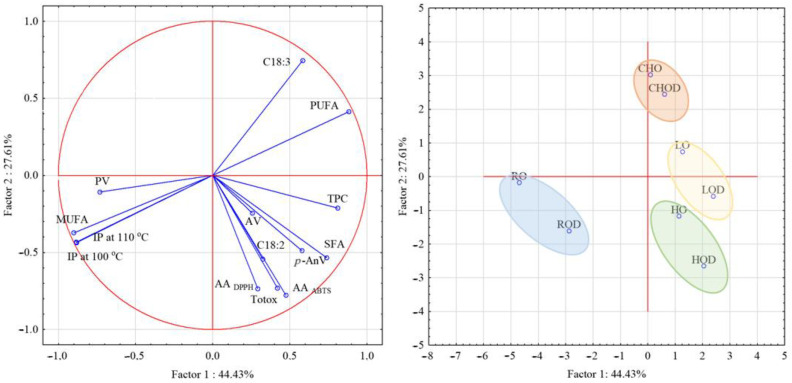
Principal component analysis (PCA) for oils without and with mullein extract, based on oils’ chemical composition data.

**Figure 3 foods-12-02391-f003:**
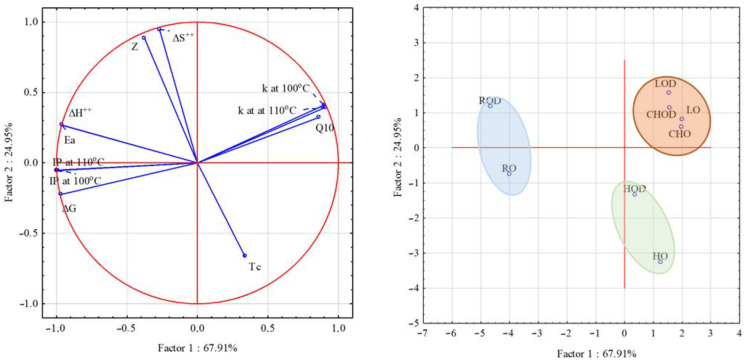
Principal component analysis (PCA) for oils without and with mullein extract, based on oils’ kinetic–thermodynamic data.

**Table 1 foods-12-02391-t001:** Induction time and protection factor of cold-pressed oils and oils with mullein extracts at 110 °C (rapeseed oil) and at 90 °C (hempseed, linseed, and chia seed oil).

Extract Concentration (mg/kg oil)	Induction Time (h)				Protection Factor (PF)			
	ROM	HOM	LOM	CHOM	ROM	HOM	LOM	CHOM
0	9.81 ^a^ ± 0.16	12.11 ^a^ ± 0.11	6.70 ^a^ ± 0.04	6.44 ^a^ ± 0.14	-	-	-	-
2	9.82 ^ab^ ± 0.11	13.78 ^b^ ± 0.34	6.71 ^a^ ± 0.04	6.55 ^ab^ ± 0.04	1.00	1.14	1.00	1.02
5	9.98 ^ab^ ± 0.05	13.84 ^b^ ± 0.15	6.72 ^ab^ ± 0.17	6.65 ^ab^ ± 0.10	1.02	1.14	1.00	1.03
10	10.01 ^bcd^ ± 0.08	13.91 ^b^ ± 0.18	6.72 ^ab^ ± 0.10	6.82 ^ab^ ± 0.05	1.02	1.15	1.00	1.06
15	10.02 ^abc^ ± 0.06	14.05 ^b^ ± 0.41	6.75 ^ab^ ± 0.19	6.80 ^ab^ ± 0.14	1.02	1.16	1.01	1.06
20	10.32 ^abcd^ ± 0.11	13.74 ^b^ ± 0.50	7.17 ^b^ ± 0.03	6.96 ^b^ ± 0.20	1.05	1.13	1.07	1.08
30	10.35 ^abcd^ ± 0.05	13.75 ^b^ ± 0.54	7.19 ^b^ ± 0.06	6.73 ^ab^ ± 0.26	1.05	1.14	1.07	1.04
40	10.42 ^abcd^ ± 0.03	13.57 ^ab^ ± 0.17	7.20 ^b^ ± 0.06	6.82 ^ab^ ± 0.04	1.06	1.12	1.08	1.06
60	10.79 ^d^ ± 0.37	13.65 ^b^ ± 0.44	7.25 ^b^ ± 0.25	6.87 ^ab^ ± 0.03	1.10	1.13	1.08	1.07
100	10.44 ^bcd^ ± 0.23	13.51 ^b^ ± 0.21	6.95 ^ab^ ± 0.16	6.85 ^ab^ ± 0.03	1.06	1.12	1.04	1.06
200	10.62 ^cd^ ± 0.11	13.71 ^b^ ± 0.03	6.99 ^ab^ ± 0.23	6.86 ^ab^ ± 0.11	1.08	1.13	1.04	1.06

ROM—rapeseed oil with extract; HOM—hempseed oil with extract: LOM—linseed oil with extract; CHOM—chia oil with extract; ^a–d^—letters show a statistically significant difference in the column at a *p*-value ≤ 0.05.

**Table 2 foods-12-02391-t002:** Oils’ quality features before and after the addition of mullein extracts.

Oil	AV (mg KOH/g)	PV (mEq O_2_/kg)	*p*-AnV	Totox (2PV + *p*-AnV)
RO	1.22 ^bc^ ± 0.01	1.74 ^b^ ± 0.06	0.37 ^a^ ± 0.14	3.85 ^b^ ± 0.03
HO	1.01 ^a^ ± 0.01	6.30 ^c^ ± 0.14	0.74 ^ab^ ± 0.30	13.34 ^d^ ± 0.59
LO	1.67 ^d^ ± 0.02	1.98 ^b^ ± 0.03	0.62 ^ab^ ± 0.09	4.58 ^b^ ± 0.15
CHO	1.11 ^ab^ ± 0.02	0.77 ^a^ ± 0.04	0.64 ^ab^ ± 0.07	2.18 ^a^ ± 0.15
ROM (60 mg/kg)	1.62 ^d^ ± 0.02	1.74 ^b^ ± 0.01	0.61 ^ab^ ± 0.13	4.09 ^b^ ± 0.10
HOM (15 mg/kg)	1.33 ^c^ ± 0.01	10.50 ^d^ ± 0.21	1.20 ^bc^ ± 0.20	22.20 ^e^ ± 0.75
LOM (60 mg/kg)	2.23 ^e^ ± 0.08	2.00 ^b^ ± 0.10	1.71 ^c^ ± 0.08	5.71 ^c^ ± 0.20
CHOM (20 mg/kg)	1.19 ^b^ ± 0.01	0.77 ^a^ ± 0.01	0.19 ^a^ ± 0.02	1.73 ^a^ ± 0.01

RO—rapeseed oil; HO—hempseed oil; LO—linseed oil; CHO—chia seed oil; ROM—rapeseed oil with extract; HOM—hempseed oil with extract: LOM—linseed oil with extract; CHOM—chia oil with extract; AV—acid value; PV—peroxide value; *p*-AnV—*p*-anisidine value; Totox—total oxidative stability. ^a–e^—letters show a statistically significant difference in the column at a *p*-value ≤ 0.05.

**Table 3 foods-12-02391-t003:** Oils’ fatty acids composition changes after mullein extract addition (%).

Fatty Acids (%)					Oils			
	RO	ROM (60 mg/kg)	HO	HOM (15 mg/kg)	LO	LOM (60 mg/kg)	CHO	CHOM [20 mg/kg]
C16:0	2.54 ^a^	2.36 ^a^	2.86 ^ab^	3.09 ^ab^	3.02 ^ab^	3.42 ^b^	3.06 ^ab^	3.18 ^ab^
C18:0	2.20 ^a^	1.73 ^a^	2.88 ^ab^	2.93 ^b^	4.35 ^c^	4.24 ^c^	2.57 ^ab^	2.38 ^ab^
C18:1	68.15 ^d^	69.14 ^d^	14.40 ^b^	13.85 ^b^	15.28 ^c^	14.94 ^bc^	7.47 ^a^	7.40 ^a^
C18:2	16.21 ^b^	17.06 ^b^	57.86 ^c^	55.77 ^c^	11.64 ^a^	11.15 ^a^	18.32 ^b^	17.98 ^b^
α-C18:3	7.16 ^a^	6.83 ^a^	16.06 ^b^	16.80 ^b^	65.28 ^c^	65.16 ^c^	68.34 ^c^	68.72 ^c^
Ɣ-C18:3	-	-	3.43 ^a^	3.47 ^a^	-	-	-	-
C20:0	1.26 ^b^	0.86 ^a^	0.85 ^a^	1.41 ^b^	-	-	-	-
C20:1	1.96 ^b^	1.59 ^b^	0.42 ^a^	0.63 ^a^	-	-	-	-
C22:0	-	-	0.69 ^a^	1.16 ^b^	-	-	-	-
Other	0.52	0.43	0.55	0.89	0.43	1.09	0.24	0.34
⅀SFA	6.00	5.38	7.83	9.48	7.80	8.75	5.87	5.90
⅀MUFA	70.11	70.73	14.82	14.48	15.28	14.94	7.47	7.40
⅀PUFA	23.37	23.89	77.35	76.04	76.92	76.31	86.66	86.70

RO—rapeseed oil; HO—hempseed oil; LO—linseed oil; CHO—chia seed oil; ROM—rapeseed oil with extract; HOM—hempseed oil with extract: LOM—linseed oil with extract; CHOM—chia oil with extract; SFA—saturated fatty acids; MUFA—monounsaturated fatty acids; PUFA—polyunsaturated fatty acids; ^a–d^—letters show a statistically significant difference in the row at a *p*-value ≤ 0.05.

**Table 4 foods-12-02391-t004:** Total phenolic compounds content in tested oils without and with the addition of mullein extract.

Extract Concentration [mg/kg oil]	Total Phenolic Compounds Content (mg GAE/100 g)			
	ROM	HOM	LOM	CHOM
0	235.42 ^a^ ± 1.32	385.70 ^ab^ ± 1.36	374.64 ^ab^ ± 0.54	313.70 ^a^ ± 0.78
2	246.25 ^b^ ± 1.87	386.67 ^ab^ ± 1.51	374.12 ^a^ ± 0.38	355.69 ^b^ ± 1.22
5	257.56 ^c^ ± 1.42	385.46 ^ab^ ± 1.32	374.50 ^ab^ ± 0.56	381.79 ^c^ ± 1.43
10	298.16 ^d^ ± 1.07	385.96 ^ab^ ± 1.98	387.50 ^c^ ± 1.54	391.58 ^d^ ± 1.00
15	340.75 ^e^ ± 1.45	399.50 ^c^ ± 2.01	384.97 ^b^ ± 0.93	394.76 ^e^ ± 0.75
20	351.95 ^f^ ± 1.10	405.93 ^d^ ± 1.56	383.42 ^b^ ± 1.23	401.24 ^f^ ± 0.87
30	354.28 ^g^ ± 0.98	407.36 ^e^ ± 1.32	393.25 ^d^ ± 1.05	405.99 ^g^ ± 1.43
40	358.72 ^h^ ± 1.65	410.62 ^h^ ± 1.76	391.03 ^c^ ± 0.54	419.48 ^h^ ± 1.08
60	363.25 ^i^ ± 0.54	408.50 ^i^ ± 1.07	397.24 ^d^ ± 0.97	423.33 ^i^ ± 1.54
100	364.97 ^j^ ± 0.43	411.38 ^h^ ± 1.26	418.63 ^e^ ± 1.23	428.24 ^j^ ± 1.01
200	372.18 ^k^ ± 0.87	433.78 ^j^ ± 1.46	476.96 ^f^ ± 1.54	444.10 ^k^ ± 1.23
Flowers extract	1678.0 ± 2.23			

ROM—rapeseed oil with extract; HOM—hempseed oil with extract: LOM—linseed oil with extract; CHOM—chia oil with extract; GAE—Gallic Acid Equivalent, ^a–k^—letters show a statistically significant difference in the column at a *p*-value ≤ 0.05.

**Table 5 foods-12-02391-t005:** Antioxidant activity of analyzed oils using DPPH and ABTS radicals.

Oil	ABTS		DPPH	
	AA (µM Trolox/kg)	% Reducing ABTS	AA (µM Trolox/kg)	% Reducing DPPH
RO	489.0 ^c^ ± 2.1	54.7 ^c^	36.9 ^a^ ± 5.9	15.8 ^b^
HO	804.2 ^e^ ± 5.2	71.7 ^f^	147.8 ^d^ ± 5.1	20.6 ^e^
LO	751.0 ^f^ ± 8.1	68.7 ^d^	135.5 ^c^ ± 5.5	20.4 ^d^
CHO	171.1 ^a^ ± 4.1	37.6 ^a^	25.6 ^a^ ± 5.2	16.0 ^a^
ROM (60 mg/kg)	525.0 ^d^ ± 8.5	56.6 ^e^	221.7 ^g^ ± 6.4	24.3 ^h^
HOM (15 mg/kg)	886.1 ^g^ ± 9.2	76.0 ^g^	184.9 ^f^ ± 7.3	22.3 ^g^
LOM (60 mg/kg)	888.8 ^g^ ± 10.1	76.1 ^g^	155.3 ^e^ ± 7.1	20.9 ^f^
CHOM (20 mg/kg)	324.9 ^b^ ± 7.2	46.8 ^b^	102.8 ^b^ ± 4.6	18.6 ^c^
Flowers extract	5234.2 ± 11.2	-	1060.0 ± 14.3	-

RO—rapeseed oil; HO—hempseed oil; LO—linseed oil; CHO—chia seed oil; ROM—rapeseed oil with extract; HOM—hempseed oil with extract; LOM—linseed oil with extract; CHOM—chia oil with extract; AA—antioxidant activity; ^a–h^—letters show a statistically significant difference in the column at a *p*-value ≤ 0.05.

**Table 6 foods-12-02391-t006:** Oxidative stability of tested cold-press oils with and without the addition of mullein extract at a temperature of 80–130 °C.

Oil	Induction Time (h)					
	80 °C	90 °C	100 °C	110 °C	120 °C	130 °C
RO	-	-	19.24 ^d^	9.81 ^e^	5.14 ^a^	2.64 ^a^
HO	23.98 ^d^	12.11 ^d^	6.30 ^b^	3.13 ^c^	-	-
LO	13.48 ^b^	6.70 ^b^	3.41 ^a^	1.72 ^b^	-	-
CHO	13.10 ^a^	6.44 ^a^	3.31 ^a^	1.68 ^a^	-	-
ROM (60 mg/kg)	-	-	21.32 ^e^	10.79 ^f^	5.50 ^b^	2.83 ^b^
HOM (15 mg/kg)	28.01 ^e^	14.05 ^e^	6.80 ^c^	3.60 ^d^	-	-
LOM (60 mg/kg)	14.31 ^c^	7.25 ^b^	3.44 ^a^	1.81 ^b^	-	-
CHOM (20 mg/kg)	13.89 ^b^	6.96 ^c^	3.32 ^a^	1.78 ^b^	-	-

RO—rapeseed oil; HO—hempseed oil; LO—linseed oil; CHO—chia seed oil; ROM—rapeseed oil with extract; HOM—hempseed oil with extract: LOM—linseed oil with extract; CHOM—chia oil with extract; ^a–f^—letters show a statistically significant difference in the column at a *p*-value ≤ 0.05.

**Table 7 foods-12-02391-t007:** Kinetic oxidation parameters analyzed oils based on Rancimat apparatus results.

Oil	Oxidation Kinetics Parameters															
	Ea (kJ/mol)	T_c_ (K^−1^)	Z (h^−1^)				*k* (h^−1^) at			ΔH^++^ (kJ/mol)	ΔS^++^ (J/mol K)	Q10	ΔG at 100 °C (kJ/mol)	ΔG at 110 °C (kJ/mol)	IP^25 °C^ (days)	IP^4 °C^ (days)
				80 °C	90 °C	100 °C	110 °C	120 °C	130 °C							
RO	82.48 ^ab^	6.88 × 10^2^	1.85 × 10^13^	-	-	0.0520 ^aA^	0.1019 ^aB^	0.1946 ^aC^	0.3788 ^aD^	79.3 ^ab^	−127.0 ^ab^	1.81	126.67 ^b^	127.94 ^b^	40	137
HO	76.06 ^a^	7.34 × 10^2^	7.35 × 10^12^	0.0417 ^abA^	0.0818 ^aB^	0.1587 ^bC^	0.3195 ^cD^	-	-	73.0 ^a^	−134.2 ^b^	1.84	123.06 ^ab^	124.40 ^ab^	16	58
LO	76.99 ^ab^	6.85 × 10^2^	1.81 × 10^13^	0.0741 ^abA^	0.1493 ^aB^	0.2933 ^cC^	0.5814 ^deD^	-	-	74.0 ^ab^	−126.6 ^ab^	1.86	121.22 ^ab^	122.49 ^ab^	10	35
CHO	76.73 ^a^	6.83 × 10^2^	1.71 × 10^13^	0.0763 ^bA^	0.1553 ^bB^	0.3021 ^cC^	0.5952 ^eD^	-	-	73.7 ^a^	−127.1 ^ab^	1.85	121.11 ^ab^	122.38 ^ab^	9	33
ROM (60 mg/kg)	84.11 ^b^	6.73 × 10^2^	2.79 × 10^13^	-	-	0.0469 ^aA^	0.0927 ^aB^	0.1818 ^aC^	0.3534 ^aD^	80.9 ^b^	−123.5 ^ab^	1.83	126.97 ^b^	128.20 ^b^	47	167
HOM (15 mg/kg)	77.34 ^ab^	6.77 × 10^2^	9.83 × 10^12^	0.0357 ^aA^	0.0712 ^aB^	0.1471 ^bC^	0.2778 ^bD^	-	-	74.5 ^ab^	−131.8 ^b^	1.85	123.66 ^ab^	124.98 ^ab^	19	71
LOM (60 mg/kg)	78.09 ^ab^	6.95 × 10^2^	2.47 × 10^13^	0.0699 ^abA^	0.1379 ^aB^	0.2907 ^cC^	0.5525 ^dD^	-	-	75.0 ^ab^	−124.1 ^ab^	1.86	121.29 ^ab^	122.53 ^ab^	10	37
CHOM (20 mg/kg)	77.61 ^ab^	6.90 × 10^2^	2.17 ×10^13^	0.0720 ^abA^	0.1437 ^bB^	0.3012 ^cC^	0.5618 ^deD^	-	-	74.6 ^ab^	−125.2 ^ab^	1.85	121.30 ^ab^	122.55 ^ab^	10	35

RO—rapeseed oil; HO—hempseed oil; LO—linseed oil; CHO—chia seed oil; ROM—rapeseed oil with extract; HOM—hempseed oil with extract: LOM—linseed oil with extract; CHOM—chia oil with extract; AA—antioxidant activity; Ea—activation energy; Tc—temperature coefficient; Z—pre-exponential factor; k—constant reaction rate; ΔH^++^—enthalpy; ΔS^++^—entropy; ΔG—free enthalpy of reaction; IP—induction period; ^a–e^ letters show a statistically significant difference in the column; ^A–D^—letters show a statistically significant difference in the row at a *p*-value ≤ 0.001.

## Data Availability

The data used to support the findings of this study can be made available by the corresponding author upon request.

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
