# Peer review of "Effect of the Freeze-Dried Mullein Flower Extract (Verbascum nigrum L.) Addition on Oxidative Stability and Antioxidant Activity of Selected Cold-Pressed Oils"

_foods, 2023, doi:10.3390/foods12122391_

Round 1

Reviewer 1 Report (New Reviewer)

This manuscript is on the preparation of mullein flower extract and its incorporation with some cold press oil to assay its antioxidative properties. The subject of study is interesting, but it needs some revision. Title could be revised to "Effect of mullein (Verbascum nigrum L.) flower extract addition on oxidative stability of selected cold-pressed oils.

Abstract should be informative presenting some data, for example how much was the the best stability?

In introduction part it is better to separate the different texts owing the different subjects for example text related to the flower should be separated from antioxidant part, from line 52 should be separate paragraph.

In the method section, it is not clear that finally how the extract was added to the oils?

In the results section, there are data on the Peroxide and acid values, but there is no method of their determination.

In Tables, there are abbreviation that not defined. Tables are independent from the text.

Table 2 shows that the extract has poroxidative effect and can increase the PV and other parameters, but in the text there is another interpretation.

If the extract increase the PV, how it can stabilize the oil and give the lower induction period in rancimat? THis should be carefully discussed and clarified in the manuscript.

Author Response

Dear Reviewer,

Thank you for your contribution to improving my manuscript. I hope that the changes made meet your expectations. Below are responses to your comments:

Comment 1:

Abstract should be informative presenting some data, for example how much was the the best stability?

Answer:

Text has been added accordingly. However, due to the limited number of words, the most important result was referred to.

Abstract: The aim of the study was to analyze the influence of mullein flower extract addition on the oxidative stability and antioxidant activity of cold-pressed oils with a high content of unsaturated fatty acids. The conducted research has shown that the addition of mullein flower extract increases the oxidative stability of oils, but its addition depends on the type of oil and should be selected experimentally. The extract from hemp oil exhibited the highest antioxidant properties, as evidenced by an increase in the induction time at 90 °C from 12.11 hours to 14.05 hours. Additionally, the extract demonstrated a protective factor of 1.16. Oils (rapeseed, chia seed, linseed, hemp seed) without and with the addition of mullein extract (2-200 mg of extract/kg of oil) were analyzed for oxidative stability, phenolic compounds content and antioxidant activity using DPPH and ABTS•+ radicals. The kinetics parameters were calculated based on the oils' oxidative stability results. In rapeseed and linseed oil, the best stability was found for samples with 60 mg of extract/kg of oil, while in chia seed oil and hemp oil, 20 and 15 mg of extract/kg of oil, respectively. The addition of the extract statistically significantly increased the antioxidant activity of oils and the total content of phenols. The extract increased the activation energy (Ea) and decreased the constant oxidation rate (k).

Comment 2:

In introduction part it is better to separate the different texts owing the different subjects for example text related to the flower should be separated from antioxidant part, from line 52 should be separate paragraph.

Answer:

Separated and a new paragraph has been created.

Comment 3:

In the method section, it is not clear that finally how the extract was added to the oils?

Answer:

Text has been added.

The test samples were prepared by mixing each of the oils with different concentrations of alcoholic extract, in the form of a syrup (Figure 1) (2, 5, 10, 15, 20, 30, 40, 60, 100, and 200 mg of extract per 1 kg of oil), as well as fresh oil without the addition of extract.

Comment 4:

In the results section, there are data on the Peroxide and acid values, but there is no method of their determination.

Answer:

Methodologies for peroxide acid value are presented, quoted in section 2.3.

“2.3. Oils' Quality Assessment

Fatty values, including acid value (AV), peroxide value (PV), and p-anisidine value (p-AnV), were analyzed to assess the initial quality of the oils and their quality after the addition of mullein extracts. The methods used for analysis were AOCS Cd 3d-63 [15] for AV, AOCS Cd 8-53 [16] for PV, and AOCS Cd 18-90 [17] for p-AnV. The total oxidation value, also known as the TOTOX indicator, was calculated using the formula: TOTOX indicator = (2 × PV) + p-AnV.”

Comment 5:

In Tables, there are abbreviation that not defined. Tables are independent from the text.

Answer:

The abbreviations have been added under the Tables.

Comment 6:

Table 2 shows that the extract has poroxidative effect and can increase the PV and other parameters, but in the text there is another interpretation.

Answer:

 An increase in PV was noted only for hemp oil, in other cases the peroxide value did not differ statistically significantly.

Here is a snippet describing the results:

"Rapeseed, linseed, and chia seed oils showed similar peroxide values with and without the extract, and there was no statistically significant difference between them at a signifi-cance level of p-value ≤ 0.05. However, in the case of hemp oil, the obtained results differed significantly, which could be attributed to the high content of chlorophylls in hemp oil, which can act as prooxidants upon exposure to light. Phenolic compounds present in the extract may have reacted with radicals and formed peroxide structures during analysis, leading to higher titration results [3].”

Comment 7:

If the extract increase the PV, how it can stabilize the oil and give the lower induction period in rancimat? This should be carefully discussed and clarified in the manuscript.

Answer:

The peroxide value is not the only differentiator that affects oil stability, so it's hard to compare to stability. The process of oil oxidation is complex and depends on many factors. Especially at high temperatures, in which the oxidation process was carried out. The oil may have a lower peroxide value and a longer induction time because the oxidation also depends on the content of various pro-oxidant and anti-oxidant compounds. Which was confirmed by the PCA analysis.

“As shown in the presented biplot (Figure 2), the induction time of oils depended mainly on their fatty acid composition, PUFA and MUFA content, and to a lesser extent on the peroxide value and total content of phenolic compounds.”

Text has been added to the section – “3.2. Impact of the extract addition on the basic distinguishing features of oil quality”:

Furthermore, despite an increase in peroxide value following the addition of the antioxidant, the overall stability of the oils improved. This can be attributed to the fact that at higher temperatures, such as 80 °C, the formation of hydroperoxides occurs alongside their decomposition at a notable rate. These hydroperoxides break down into both volatile and non-volatile compounds, with the Rancimat apparatus focusing on volatile oxidation products to determine the outcome.

Reviewer 2 Report (Previous Reviewer 2)

The reviewed version seems to be a significant improvement, though is quite difficult to read the reviewed text with so many coloured/ deleted words.

However, please do consider a former issue:

#2.4 - add relevant details about the stated there-stage extraction performed to obtain the alcoholic extract – STAGE 1 WAS…/ STAGE 2 WAS… STAGE 3 WAS…; besides, since you added in L.184 the mention “with continuous agitation”, add the mixer type/producer/ number of rotations/ min

For me, the added figure 1 has no scientific value, but more an aesthetic one; you can consider deleting it (it will cause also some printing difficulties, since it is a colour image and it needs to have a high resolution in order to look good in the printed version).

Only minor editing of English language is required

Author Response

Dear Reviewer,

Thank you for your contribution to improving my manuscript. I hope that the changes made meet your expectations. Below are responses to your comments:

Comment 1:

#2.4 - add relevant details about the stated there-stage extraction performed to obtain the alcoholic extract – STAGE 1 WAS…/ STAGE 2 WAS… STAGE 3 WAS…; besides, since you added in L.184 the mention “with continuous agitation”, add the mixer type/producer/ number of rotations/ min

Answer:

I think we used bad word, it was 3-times extraction not a 3-step.

The methodology has been changed accordingly:

“The ethanolic extract of mullein flowers was obtained through three steps of alcoholic extraction. Initially, freeze-dried mullein flowers were ground using a mortar. The first step involved extracting 2 grams of flowers with 40 mL of 60% ethanol at 60 °C for 1 hour in a water bath with shaking (Vibra 6, AJL, Cracow, Poland). Following the extraction, the sample was centrifuged for 5 minutes at 5000 rpm using a centrifuge (MPW-352, MPW, PoznaÅ„, Poland). The resulting extract was transferred to another vessel, and the residue underwent a second extraction step using 30 mL of 60% ethanol, following the same parameters and post-treatment as the first stage. The third extraction step utilized 30 mL of solvent, with the procedure remaining consistent with the first and second extraction steps. Once the extractions were complete, the extracts were collected, filtered, and evaporated in a vacuum evaporator (Rotavapor R-210, Buchi, Flawil, Sankt Gallen, Switzerland) at 60 °C, resulting in a final volume of 30 mL for the ethanolic extract.”

Comment 2:

For me, the added figure 1 has no scientific value, but more an aesthetic one; you can consider deleting it (it will cause also some printing difficulties, since it is a colour image and it needs to have a high resolution in order to look good in the printed version).

Answer:

I leave that decision to the editor because one of the previous reviewers wanted to add pictures of flowers and extract.

Round 2

Reviewer 1 Report (New Reviewer)

Manuscript was not revised according to the comments and still there are many concerns on the writing and scientific quality of the revised manuscript. Even in the added text, there are some scientific mistakes. For example: in Abstract, line 20: "The extract from hemp oil exhibited...", there is no extract of hemp seed oil which has been written in the abstract. Also, there is no references in the added text after revision. Please check carefully the previous comments and include them in the whole manuscript.

There are many scientific errors which were stated in the previous comments and this time as well which needs serious attention to get the scientific manuscript to be publishable in this journal.

Author Response

Dear reviewer,

Answers to your comments are included in the attached file. I hope that this time they will meet your requirements, if anything needs to be improved, please comment.

This manuscript is a resubmission of an earlier submission. The following is a list of the peer review reports and author responses from that submission.

Round 1

Reviewer 1 Report

In this manuscript the effect of adding ethanolic extracts of mullein flower to four oils, on the oxidation-related parameters, phenolic content, and antioxidant activity in vitro was investigated.

The subject of the manuscript is interesting, taking into account the trends towards natural antioxidants. However, the document still needs a deeper analysis of the obtained data, as well as some additional data, in order to improve its scientific quality.

Section 2.1.

If possible, indicate the temperature of the oil at the press outlet.

What is the concentration of the ethanolic extracts? Provide all the characterization data of the extract: Total phenolic content, antioxidant activity, etc.

Sections 2.2, 2.3, 2.6, 2.7, and 2.8

Indicate how long after the samples were prepared, these analyzes were made.

Section 3

The discussion of all the results should be associated to the fatty acid composition of each one of the oils.

Explain the trend of the data presented in Table 1. All the results show that both the induction time and the protection factor slightly increase as the concentration of the extract increase, and then they  trend to decrease when the concentration of the extract is high.

Section 3.2.

Explain the reason of the increase in the AV of all oils after adding the extracts. It is not enough only to compare the data with previous results published in the available literature.

Explain the reason of the increase in the PV of the hemp oil sample after adding the extracts. It is not enough only to compare the data with previous results published in the available literature.

Explain the variation of the p-An values of the oil samples after adding the extracts. In the case of three oils (RO, HO, and LO), the p-An value significantly increased after the addition of the extract, but in the case of CHO, the p-An value significantly decreased after the addition of the extract. What could explain this behavior?

Section 3.3

The fatty acid composition of the oils should be correlated with all the other obtained data.

Example: Why the HO having higher concentration of PUFA displyed higher induction time than RO which is mainly a MUFA oil?

Section 3.4

In Table 4 add the values of the standard deviation of the data.

In Table 4, according to the data presented, the highest content of phenolics corresponds to the HO, not to the LO, as mentioned in the text.

What is the concentration of phenolics in the ethanolic extracts added to the oils?

What is the antioxidant capacity of the ethanolic extracts added to the oils?

In Table 5 add the values of the standard deviation of the data.

It would have been interesting to study the samples during a certain storage time.

Author Response

Dear Reviewer,

Thank you for taking the time to review our article, "Effect of the freeze-dried mullein flower extract (Verbascum L.) addition on oxidative stability and antioxidant activity of selected cold-pressed oils". We appreciate your thoughtful comments and suggestions, and we have carefully considered each of the points you raised.

The answers were sent in a file.

We hope that these revisions have addressed your concerns and improved the quality of our manuscript. Thank you again for your valuable feedback.

Sincerely,

Edyta Symoniuk

Reviewer 2 Report

 The objective of the study described in this manuscript was to investigate the effect of adding mullein flower extracts on the antioxidant properties and oxidative stability of cold-pressed oils with a high content of polyunsaturated fatty acids.

Unfortunately the manuscript has several draw-backs and needs a revision, hence please consider the following suggestions for an improved version:

- English needs improvement in some instances (try to break long phrases in shorter ones);

- L.70 – consider deleting “while also utilising the medical properties of mullein” unless you can prove that the added amounts contribute indeed to certain health effects

#2 – add a new paragraph devoted to materials used in this research (reagents, standards, carrier gasses) and mention type/quality/supplier;

#2.1 – ask for relevant technological parameters from the manufacturers and include them in your manuscript, as well as the storage temperature for oils; without these the experimental data are impossible to replicate;

- L. 107, 109, 137, 138 – replace “ml”> mL;

#2.2 – add relevant details about the application of the stated methods in this context (sample processing, sample amounts, reagents, equipment, etc.)

#2.4 - add relevant details about the stated there-stage extraction performed to obtain the alcoholic extract;

- L.135 and 139 – “The absorbance of the reaction mixture was measured at λ = 734 nm” appears twice – delete one instance and re-phrase accordingly;

- Table 7 – define Q10 and discuss the obtained values;

- L.431, 434 - rephrase, replacing “optimal” since the experimental design of this study was not directed towards optimization

Author Response

(The authors gave the same response as above.)

Reviewer 3 Report

Very well work, but you wrote in the manuscript title "mullein flower extract (Verbascum L.)" without describing which species? 

In the introduction you need to highlight about the active ingredient in the Mullein, like saponins (https://doi.org/10.1016/j.foodhyd.2018.02.050), etc., 

In materials and methods the flower harvested in which stage? I prefer to add photo for the flower and the extracts

Author Response

(The authors gave the same response as above.)
